# Detection of a Novel Papillomavirus Type within a Feline Cutaneous Basal Cell Carcinoma

**DOI:** 10.3390/vetsci9120671

**Published:** 2022-12-01

**Authors:** John S. Munday, Hayley Hunt, Geoff Orbell, Hayley Pfeffer

**Affiliations:** 1School of Veterinary Science, Massey University, Palmerston North 4410, New Zealand; 2Gribbles Veterinary Pathology Ltd., Palmerston North 4410, New Zealand; 3Vetora, Te Awamutu 3800, New Zealand

**Keywords:** cat, papillomavirus, neoplasia, cancer, basal cell carcinoma, skin, carcinogenesis

## Abstract

**Simple Summary:**

Basal cell carcinomas (BCCs) are rare skin tumors of cats. The presently described cat developed a BCC on the flank and another on the thorax. The flank BCC was large and recurred after surgery. The neoplasm contained unusual histological features that included a spindle-shaped appearance to the cells as well as the presence of numerous large cytoplasmic bodies. The cytoplasmic bodies were consistent with papillomavirus (PV)-induced cell changes and PV DNA sequences were amplified from the mass using PCR. The sequences have not been previously reported and are most likely from a novel PV type. In contrast, the thoracic BCC had a more typical histological appearance and contained no viral cytopathic changes or amplifiable PV DNA. Observations from this case add to the variety of skin lesions associated with PVs in cats as well as increasing the likely number of PV types that infect domestic cats.

**Abstract:**

A 4 cm diameter exophytic mass was excised from the left flank of a 10-year-old domestic short-haired cat. Histology of the superficial aspects of the mass revealed epidermal cells arranged in nests and trabeculae while the deeper parts of the mass consisted of small round cells arranged in sheets or bundles of elongate spindle-shaped cells. A diagnosis of basal cell carcinoma (BCC) was made. Approximately 40% of the cells throughout the neoplasm contained prominent papillomaviral (PV)-induced cell changes. The BCC recurred three months after excision and grew rapidly. At this time a smaller mass was observed on the thorax. Due to the rapid recurrence of the BCC, the cat was euthanatized. As in the initial mass, histology of the recurrent mass revealed pleomorphic cells that often contained PV-induced cell changes. In contrast, the thoracic mass appeared as a more typical BCC and contained no histological evidence of PV infection. A novel PV DNA sequence was amplified from the flank BCC. While the sequence was most (75.1%) similar to Felis catus papillomavirus (FcaPV) 6, the level of similarity between the sequences is consistent with a novel PV type. No PV DNA was amplifiable from the thoracic mass. The case is unique due to the histological features of the BCC and the presence of a putative novel PV type. Observations from the present case add to the number of PV types associated with disease in cats as well as increasing the spectrum of PV-induced lesions in this species.

## 1. Introduction

Papillomaviruses (PVs) are small, circular, double-stranded DNA viruses. PVs cause self-resolving hyperplastic warts in many species [1]. Additionally, they also cause around 5% of human cancers, as well as being associated with neoplasia in some domestic species including cats, dogs and horses [2,3]. Papillomaviruses are classified using the *open reading frame (ORF) L1* with PVs of different types having less than 90% nucleotide similarity, and PVs of different genera having less than 60% similarity [4]. 

Six different Felis catus papillomavirus (FcaPV) types have been fully sequenced [5]. FcaPV1 is classified within the *Lambdapapillomavirus* genus and is thought to cause oral papillomas. The most well-studied PV to infect domestic cats is FcaPV2. This PV type is a *Dyothetapapillomavirus*, and is thought to cause most feline viral plaques (a term that now includes the more severe counterpart Bowenoid in situ carcinomas), cutaneous squamous cell carcinomas (SCCs) and some less common skin cancer types such as basal cell carcinomas (BCCs) and Merkel cell carcinomas [6,7,8,9,10,11]. Three FcaPV types, FcaPV3, FcaPV4, and FcaPV5, are closely related and are classified in the *Taupapillomavirus* genus. These PVs are also associated with cutaneous neoplasia including SCCs and BCCs [5,12]. FcaPV6 has only been detected once when it was amplified from a cutaneous SCC [13]. 

The expression of PV proteins within a cell can cause histologically detectible PV-induced cell changes. In cats, the PV type present within a lesion can often be predicted by the appearance of these cell changes [5]. Infection of a cell by FcaPV1 can result in prominent eosinophilic roughly spherical intracytoplasmic bodies [14]. Cells infected by FcaPV2 can have cytoplasm that is expanded by clear or slightly granular blue-grey material while FcaPV3 infection can result in slender elongated perinuclear basophilic bodies [15]. While few cases have been reported, cells infected by FcaPV4 and FcaPV5 may have cytoplasm expanded by dark blue-grey material, with this change more prominent in the superficial epidermis for FcaPV4 and within deeper parts of the hair follicle for FcaPV5 [16,17]. The single SCC in which FcaPV6 DNA was detected did not demonstrate any PV-induced cell changes [13]. 

Herein is described a cat that developed a large BCC on the flank. This BCC contained a highly unusual population of round to spindle-shaped neoplastic cells. A high proportion of cells contained prominent PV-induced cell changes including cytoplasmic bodies that obscured the cell nucleus. PCR revealed the presence of DNA sequences from a putative novel PV type. 

## 2. Case Presentation and Diagnosis

A 10-year-old silver tabby domestic short-haired cat presented with a 4 cm diameter mass on the dorsal left flank. The mass had been initially observed three weeks previously and had steadily increased in size. Over the last three days, the mass had become covered by a thick crust with cloudy flocculent material observed discharging from the mass. The cat appeared otherwise healthy. Due to the possibility of a bacterial cause of the lesion, the cat was initially treated with a 7-day course of amoxicillin/clavulanic acid antibiotics at a dose of 12.5 mg/kg (Clavaseptin, Vetoquinol New Zealand, Takapuna, New Zealand).

The mass continued to enlarge over the following seven days and surgical excision was performed. During excision, the serocellular crust was removed revealing numerous cavities containing cloudy floccular material (Figure 1). The mass was fixed in 10% neutral buffered formalin for around 24 h and processed for histology. Sections were prepared using routine methods and stained using hematoxylin and eosin. 

Histology of the superficial areas of the mass revealed cells arranged in nests and thick cords extending into the underlying dermis (Figure 2). The neoplastic cells were surrounded, and separated by, large quantities of dermal fibrosis. The cells were polygonal and basophilic, and there was no evidence of keratinization. The surrounding epidermis was within normal limits with a sudden transition from normal to neoplastic epidermis. Cells within deeper areas of the mass were arranged in densely packed sheets or bundles of neoplastic cells. These cells were separated by thick areas of dense fibrous tissue and neoplastic cells often surrounded cystic spaces that contained pink amorphous material that was admixed with small quantities of cellular debris. This material was interpreted to be the floccular fluid that was observed on gross examination of the mass. The appearance of the neoplastic cells was variable within the mass. Within the superficial aspects, the neoplastic cells were large and polygonal with well-defined cell borders, large quantities of cytoplasm and a prominent round central nucleus. In contrast, within deeper areas of the neoplasm the cells ranged from small polygonal to elongated spindle-shaped cells. These cells had indistinct cell borders with variable amounts of cytoplasm and either round or elongate nuclei. Throughout the neoplasm approximately 40% of the cells contained prominent basophilic cytoplasmic bodies. These bodies often partially, or fully surrounded, the nucleus and the cytoplasmic bodies completely obscured the nucleus in some cells (Figure 3). In addition, smaller numbers of cells in which the cytoplasm was expanded by a clear space or grey-blue cytoplasm were visible throughout the neoplasm. 

Due to the histological evidence of PV infection, total DNA was extracted from a formalin-fixed paraffin-embedded tissue scroll of the lesion (NucleoSpin DNA FFPE XS kit, Macherey-Nagel, Düren, Germany). PV DNA was amplified using the MY09/11 and CP4/5 consensus primers as previously described [18]. DNA extracted from a canine oral mass that contained CPV17 was used as a positive control while no template DNA was added to the negative controls. Papillomaviral DNA was amplified using the MY09/11 and CP4/5 primers. Sequencing of the MY09/11 amplicon produced a 321 bp section of the PV *ORF L1* which was compared to other sequences in GenBank using BLAST (https://blast.ncbi.nlm.nih.gov/Blast.cgi (accessed on 1 September 2022)). The sequence had the highest similarity to FcaPV6 with 75.1% similarity between the two sequences. In comparison, the novel PV DNA sequence was only 54.6%, 56.2%, 62.6%, 58.1%, and 57.8% similar with the corresponding 321 bp sequences of FcaPV1, 2, 3, 4, and 5, respectively. The novel sequence had only 64.7% similarity with a short PV sequence that had previously been amplified from a feline BCC [19]. The presently described novel sequence was deposited into GenBank (accession number OP762604). 

Recurrence of the flank mass was observed approximately one month later and the mass continued to enlarge over the next three months until it was 8 cm in diameter. The mass was ulcerated with small quantities of a cloudy discharge. A fine needle aspirate of the mass was performed and resulting smears were stained with Diff Quik (a Romanowsky stain variant). The smears contained small numbers of a moderately pleomorphic population of plump spindle to polygonal cells that frequently contained large intracytoplasmic pale basophilic to brightly eosinophilic inclusions (Figure 4). Rare cells contained discrete magenta granules or clear vacuoles surrounding the nucleus. At this time, a 1 cm diameter mass was observed on the dorsal thorax of the cat. Due to recurrence and rapid growth of the flank mass, and the development of an additional mass, the owners elected euthanasia of the cat.

The cat underwent a full post-mortem examination. The mass from the flank was ulcerated and covered by a serocellular crust. The mass was ovoid with the longest dimension parallel to the overlying epidermis. The mass did not appear to be adhered to the underlying tissue and could be removed easily. Likewise, the thoracic mass did not appear invasive on gross examination. Histological examination of the large mass from the flank revealed a similar cell population to that of the initial mass with deeper parts comprising spindle-shaped cells surrounding cystic cavities that contained cell debris (Figure 5). As in the previously excised mass, around 40% of the cells within the neoplasm contained prominent basophilic intracytoplasmic bodies (Figure 6). Examination of the deeper aspects of the mass revealed only rare invasion of neoplastic cells into the thick surrounding fibrous capsule. Examination of the mass from the thorax revealed a similar population of neoplastic cells. However, no evidence of PV-induced cell changes was observed within this neoplasm. Immunohistochemistry revealed no immunoreactivity to antibodies against CK7, CK8/18, CK20, neuron specific enolase (NSE), or synaptophysin within either mass. The histological and immunohistochemical features of both tumors were consistent with a diagnosis of BCC. Histology of multiple tissues including draining lymph nodes, liver and lung from the cat did not reveal evidence of metastases or other significant lesions.

Total DNA was extracted from unfixed samples of both the BCC from the flank and the BCC from the thorax and evaluated for the presence of PV DNA as previously described. PV DNA sequences that were identical to those previously detected from the flank mass were again amplified from the recurrent BCC from the flank while no PV DNA was amplified from the thoracic BCC. 

## 3. Discussion 

The BCC from the flank contained a previously unreported PV DNA sequence. Definitive classification of a PV type is not possible without the complete PV *ORF L1* [4]. However, as the novel sequence was only 75% similar to previously reported PV sequences, a novel PV type is almost certain. The putative novel PV type has the highest similarity to FcaPV6 and is much less similar to the other previously reported FcaPV types. This suggests that both the putative novel PV and FcaPV6 may cluster in a novel PV genus. Interestingly, a feline BCC that contained sequences from a putative novel PV type has been previously reported [19]. However, the previously amplified sequence had low similarity with the presently described sequence, suggesting these sequences are from different PV types that are likely to be different genera. 

The mass that contained the putative novel PV type was classified as a BCC. Features consistent with this diagnosis include the arrangement of the neoplastic cells into trabeculae and the presence of cystic cavities within the neoplasm [20]. However, the presently described neoplasm was unusual due to the predominance of spindle-shaped cells in large areas of the neoplasm. The possibility that the neoplasm was a Merkel cell carcinoma was considered; however, this was excluded by the lack of immunoreactivity against CK20, NSE and synaptophysin [11,21]. Likewise, the absence of immunoreactivity against CK7 and CK8/18 suggests it is unlikely that the neoplasm was of either apocrine gland or follicular origin [22]. 

Adding to the unusual histological features visible in the BCC was the presence of PV-induced cell changes in around 40% of the neoplastic cells. While PV-induced cell changes have previously been reported in feline BCCs, these changes have previously been visible in only small numbers of cells within the center of cell nests [19]. In contrast, PV-induced cell changes were frequent throughout all areas of the neoplasm in the present case, including within both spindle-shaped cells and cells that had retained the more typical polygonal appearance expected within a BCC. The PV-induced cell changes suggest that PV protein expression was present in, and altering normal cell regulation of, a high proportion of the neoplastic cells. It is possible that the unusually high frequency of PV-induced cell changes was because the neoplastic cells were infected by an unusual PV type. However, it remains possible that the BCC in this case simply provided an unusually permissive environment for PV infection. 

Examination of the PV-induced cell changes within a PV-associated feline lesion often enables prediction of the causative type [5]. In the present case, not only were the changes unusually frequent, but such large eosinophilic cytoplasmic bodies have not previously been reported due to infection by any of the currently recognized FcaPV types. The unusual appearance of the PV-induced cell changes in the present case adds to the evidence that the DNA sequence was from a putative novel PV type. 

The presently reported cat developed two BCCs. The larger BCC from the flank contained spindle-shaped cells, histological evidence of PV infection, and PV DNA. In contrast, the BCC from the thorax had a more typical histological appearance and contained no molecular or histological evidence of PV infection. This suggests that the unusual spindle cell phenotype of the flank BCC developed as a result of infection by the novel PV type. However, additional cases are required before firm conclusions are possible. 

The presence of the PV-induced cell changes within the flank BCC provides convincing evidence that the putative novel PV had infected the neoplastic cells. Furthermore, the cell changes and unusual phenotype of the neoplastic cells are consistent with the PV infection altering normal cell growth and regulation. However, as PVs are often present as incidental infections [23], it is not possible to prove that the novel putative PV type caused the BCC in this case. It remains possible that the PV was present within the neoplasm simply as an ‘innocent bystander’. Furthermore, the absence of any PV DNA or PV-induced cell changes within the smaller thoracic BCC suggests that this neoplasm developed due to factors other than PV infection. Therefore, if the smaller BCC developed without any PV involvement, the larger mass could have likewise developed independently of any influence from the PV. 

Despite many studies using consensus PCR primers to evaluate the presence of PVs within feline skin cancers, this is the first time that the novel PV type has been identified [12,24,25]. The absence of this PV type in earlier studies suggests that infection by the novel putative PV type is rare in cats. Likewise, there is only a single report describing FcaPV6, the most closely related FcaPV type [13]. This suggests that both FcaPV6 and the putative novel PV type rarely infect, and rarely cause disease, in cats. 

As feline BCCs have been previously associated with FcaPV3, the possibility of co-infection by this, or other, PV types was considered. However, as both the MY09/11 and CP4/5 consensus primers consistently only amplified DNA sequences from the putative novel PV type from the neoplasms, the presence of additional PV types within the lesions appears to be unlikely. 

## 4. Conclusions

The BCC from the flank showed histological features that have not been previously described. Additionally, the BCC contained DNA sequences of a putative novel PV type. The findings in this case expand the number of FcaPV types as well as expanding the manifestations of PV disease in cats. 

## Figures and Tables

**Figure 1 vetsci-09-00671-f001:**
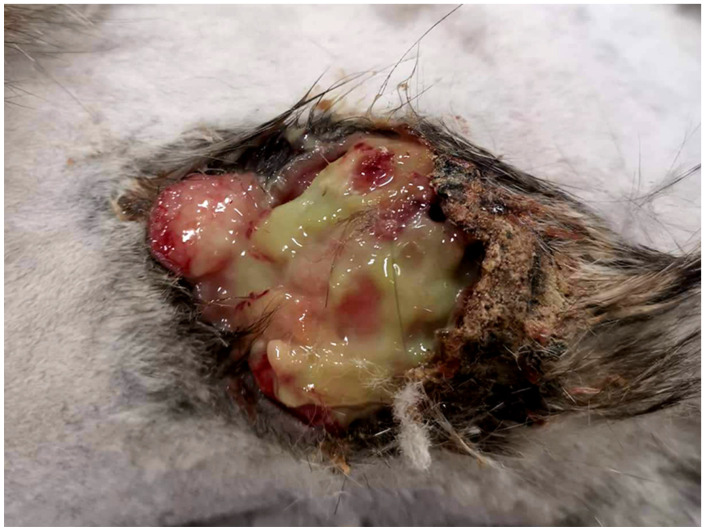
The cat presented with a 4 cm diameter mass on the flank. Removal of a serocellular crust revealed numerous cavities containing cloudy floccular material.

**Figure 2 vetsci-09-00671-f002:**
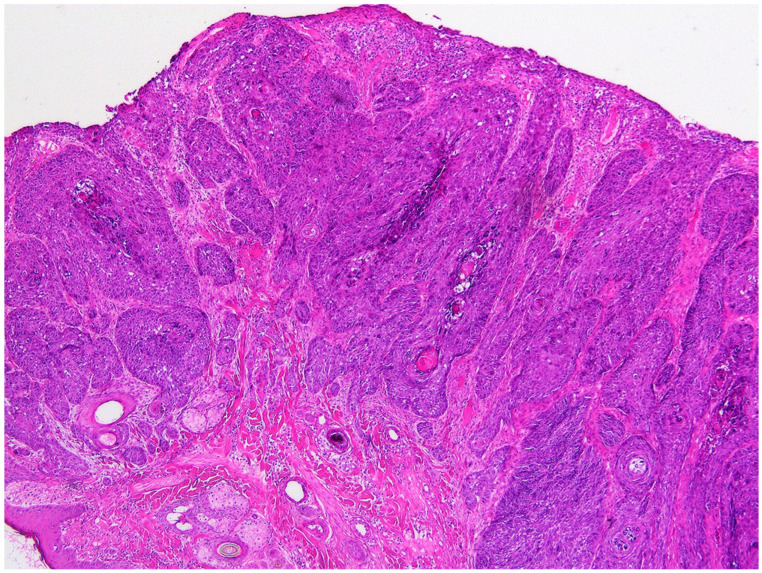
Photomicrograph of the mass from the flank. Superficial aspects of the mass consist of trabeculae of neoplastic cells extending into a dermis that is thickened by increased fibrous tissue. H&E 50×.

**Figure 3 vetsci-09-00671-f003:**
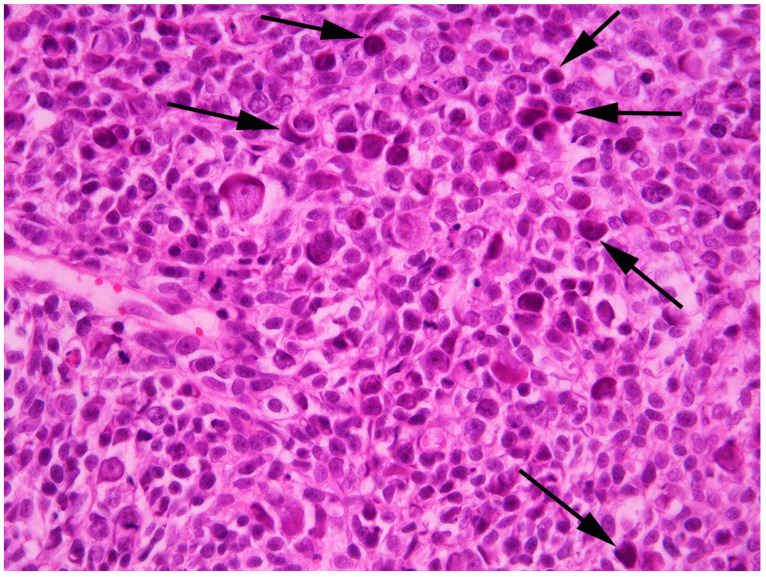
Photomicrograph of the mass from the flank. A high proportion of the cells contain large eosinophilic cytoplasmic bodies that are interpreted to be papillomavirally induced (arrows). While some remain elongate, other bodies are large and obscure the cell nucleus. H&E 400×.

**Figure 4 vetsci-09-00671-f004:**
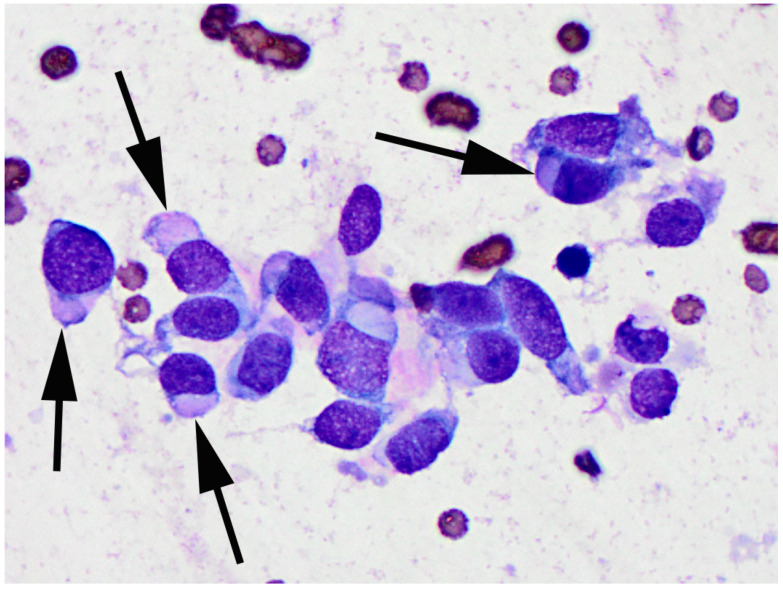
Photomicrograph of a fine-needle aspirate of the recurrent mass from the flank. Small numbers of plump spindle to polygonal cells are visible. Cells often contain large intracytoplasmic pale basophilic to brightly eosinophilic inclusions (arrows). Diff Quik 1000×.

**Figure 5 vetsci-09-00671-f005:**
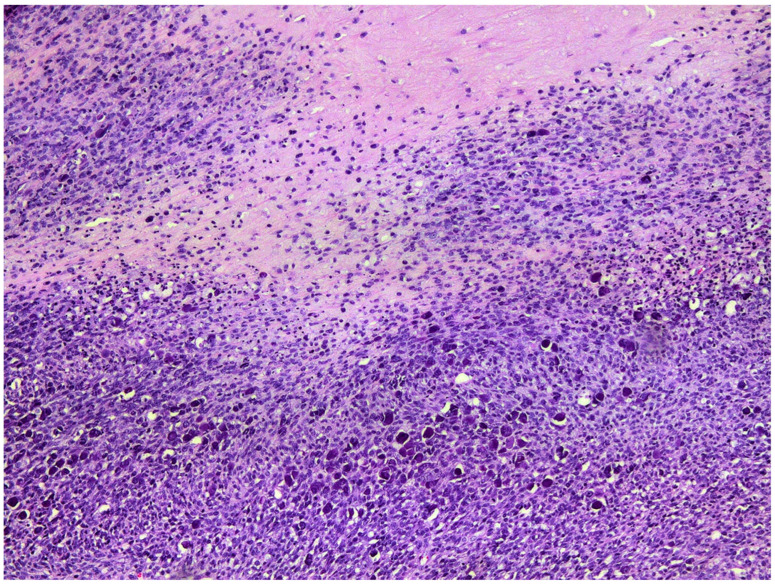
Photomicrograph of the recurrent mass from the flank. Deeper aspects of the neoplasm consist of bundles of spindle-shaped cells surrounding cystic cavities that contain eosinophilic debris. Even at this low magnification, papillomavirally induced cell changes are visible in many of the neoplastic cells H&E 100×.

**Figure 6 vetsci-09-00671-f006:**
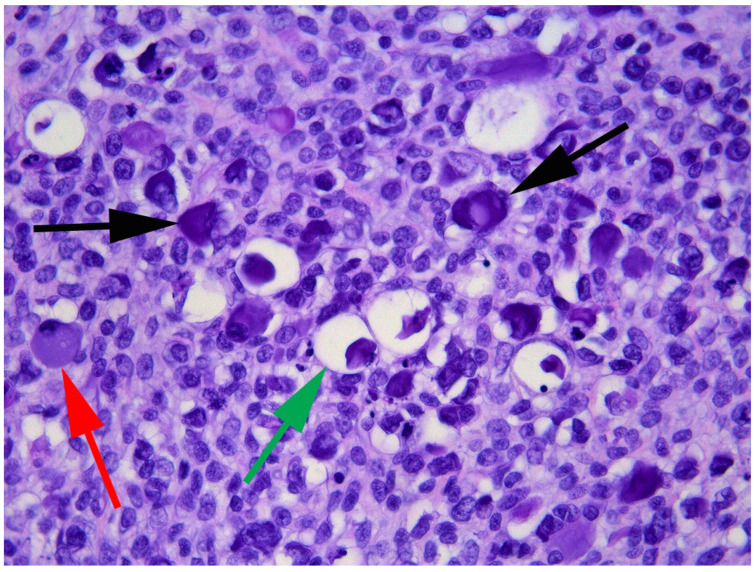
Photomicrograph of the recurrent mass from the flank. Papillomavirally induced cell changes are frequently visible including large eosinophilic cytoplasmic bodies that can obscure the cell nucleus (black arrows), cytoplasm that is expanded by blue-grey material (red arrow), and cells with cytoplasm expanded by clear material and shrunken dark nuclei (green arrow). H&E 400×.

## Data Availability

The data presented in this study are available on request from the corresponding author.

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
