# Peer review of "Detection of a Novel Papillomavirus Type within a Feline Cutaneous Basal Cell Carcinoma"

_vetsci, 2022, doi:10.3390/vetsci9120671_

Round 1

Reviewer 1 Report

Review for manuscript “Detection of a novel papillomavirus type within a feline cutaneous basal cell carcinoma.”

Thank you for inviting me to review such an interesting and well-written case report. This will certainly add to the published literature on papillomaviruses related to feline neoplasms, partly because this is a novel papillomavirus type but also due to the figures demonstrating the striking histopathological features seen, which will greatly aid pathologists in recognizing similar cases in their own diagnostic caseloads.

I only have a very few, very minor comments and suggestions to make. 

In just two places, the numeral 3 is used, when the word three might have been more appropriate (line 72 and line 126). 

Line 72: “3 weeks ago” might sound better as “three weeks previously”

Line 99: “around 40%” might sound better as “approximately 40%”

The figures are very useful and add greatly to the manuscript. There are no scale bars, but I am uncertain whether this particular journal requires scale bars or not.

Reviewer 2 Report

In this case report, the authors present a clinical case regarding a cat with papillomavirus, confirmed by histological and molecular reports. In particular, the authors report a new type of feline papillomavirus. The study is very interesting, considering the atypical spread of feline papillomavirus. The study as a whole is sufficiently well presented.
However, I would like to point out some small corrections-revisions, and some questions for clarification.

1. The author order has an order-formatting error (and should go to the end of the author list);

2. Possible error in the entry: Correspondance: ......;

3. In the simple summary, I suggest reporting very briefly, then integrating, the histological response from the thoracic mass (in order to underline the differences which, for a histopathologist and a clinician, must be immediately visible);

4. Line 42: ".... similarity, and PVs of ........". Please, correct. Also, Felis catus, in italics.

5. Line 45: "The most well-studied PV ....". Please, correct.
6. Line 46: "is a Dyothetapapillomavirus, and is thought to ...". Please, correct.
7. Line 52: ".. has only been detected once when ...". Please, correct.
8. Line 71: space at 4cm ... please correct.
9. Line 76: 7 days ...
10. Line 76: "..... course of amoxicillin / clavulanic acid antibiotic at a dose of 12.5 mg/kg (Clavaseptin, ......... New Zeland). Please correct.
Question: The weight of the cat?

Curiously, was a bacteriological examination carried out? .. No bacterial cells were identified by histological examinations ... Was the production of floccular material at the post-antibiotic treatment (at the clinical control) resolved?

11. Line 80. would I report the% dilution of formalin ... 10%? 40%?

Line 81. Here, personally, I would close paragraph 2, and with another paragraph (2.1) I would separate the part relating to the histological investigation, which starts at line 85.

Line 85: I would briefly write here the type of specific coloring used, and not report it only in the captions of the photos.

Line 87-88: "The cells were polygonal and basophilic, and no evidence of keratinization was noted." Please correct.

Line 97-98: "These cells had indistinct cell borders with variable amounts of cytoplasm and either round or elongate nuclei." Please, correct.

Regarding the discussion of viral DNA extraction and therefore the biomolecular analysis, I suggest writing it in a separate paragraph.

Considering the DNA denaturation effects of formalin, that may have affected the sequence, how long after this time, the extraction was performed after the immersion of the organ samples in formalin?

It would be useful to report this information in the paper.

Therefore, I suggest integrating it. If available, I would complement the manuscript with gel-band images. Has a hypothetical phylogenetic tree been performed?

From Line 140, it is written that the cat was subjected to an autopsy.
Line 140: "..... underwent a full necroscopy.". Please correct.

Question: what pathological conditions did lymph nodes and spleen present? Kidneys and liver?

In the discussion section,  I suggest improving and arguing better from the histological point of view the one highlighted in this study and the one reported by the study cited at n. 22.
